# Stochastic Poisson Surface Reconstruction with One Solve using Geometric Gaussian Processes

Sidhanth Holalkere [1]   David S. Bindel [1]   Silvia Sellán [2 3]   Alexander Terenin [1]

## Abstract

Poisson Surface Reconstruction is a widely-used algorithm for reconstructing a surface from an oriented point cloud. To facilitate applications where only partial surface information is available, or scanning is performed sequentially, a recent line of work proposes to incorporate uncertainty into the reconstructed surface via Gaussian process models. The resulting algorithms first perform Gaussian process interpolation, then solve a set of volumetric partial differential equations globally in space, resulting in a computationally expensive two-stage procedure. In this work, we apply recently-developed techniques from geometric Gaussian processes to combine interpolation and surface reconstruction into a single stage, requiring only one linear solve per sample. The resulting reconstructed surface samples can be queried locally in space, without the use of problem-dependent volumetric meshes or grids. These capabilities enable one to (a) perform probabilistic collision detection locally around the region of interest, (b) perform ray casting without evaluating points not on the ray's trajectory, and (c) perform next-view planning on a per-ray basis. They also do not requiring one to approximate kernel matrix inverses with diagonal matrices as part of intermediate computations, unlike prior methods. Results show that our approach provides a cleaner, more-principled, and more-flexible stochastic surface reconstruction pipeline.

## 1. Introduction

Surface reconstruction algorithms are used to process point cloud data—the most-common format in which real-world three-dimensional scans are captured—into other downstream-usable formats, such as meshes or other surface representations. Such algorithms are a key computational primitive used in computer graphics pipelines.

The most widely-used approach is *Poisson Surface Reconstruction* [17, 16]. It works by solving a partial differential equation to compute an implicit surface representation—a function where positive (resp. negative) values denote locations inside (resp. outside) the surface, with zeroes denoting the surface. Its computation relies on standard numerical techniques—most commonly the finite element method—making it simple, reliable, and efficient on compute devices well-suited to sparse linear algebra.

Real-world scans are necessarily imperfect: they involve noise, measurement error, and may even need to gracefully handle situations where parts of the object are occluded or otherwise not observed by the camera. As a consequence, surface reconstruction algorithms must both interpolate and extrapolate from noisy and potentially incomplete data as part of their computations. Poisson surface reconstruction typically does this by blurring the data using a smooth kernel: while simple and effective, this process does not quantify uncertainty about interpolation or extrapolation.

To provide a richer characterization of the information contained in a scan, a recent line of work proposes *Stochastic Poisson Surface Reconstruction* [27]—a Bayesian formalism which incorporates uncertainty about the scan through the use of Gaussian process priors. The implicit surface representation is then recovered by applying Bayes' Rule in the solution space of the respective Poisson equation's stochastic analog. This approach not only quantifies uncertainty, it provides a potential path for selecting optimal scan directions in a technically principled manner, mirroring data acquisition techniques from areas such as Bayesian optimization [11] and probabilistic numerics [13].

A key challenge in stochastic Poisson surface reconstruction is that it requires more complex computations than its non-stochastic analogs. The simplest approach, proposed by Sellán and Jacobson [27], is to first condition the Gaussian process on point-cloud observations, then solve the required Poisson equation to obtain the surface representation. Un-

Code: HTTPS://GITHUB.COM/SHOLALKERE/GEOSPSR.

[1]Cornell University [2]Columbia University [3]MIT. Correspondence to: Sidhanth Holalkere <sh844@cornell.edu>.

*Proceedings of the 42nd International Conference on Machine Learning*, Vancouver, Canada. PMLR 267, 2025. Copyright 2025 by the author(s).

fortunately, this approach suffers from the classically-cubic scalability of Gaussian processes: the authors handle this by approximating kernel matrix inverses with diagonal matrices. Then, finite elements are used to solve the Poisson equation—a potentially expensive operation in its own right due to reliance on a volumetric mesh or grid.

In this work, we study the following question: *can one compute the posterior of the implicit surface directly, without solving separate linear systems for interpolating the data and calculating the implicit surface?* We answer this affirmatively using methods from inter-domain Gaussian processes, geometric Gaussian processes, and related areas. Our techniques:

1. Produce posterior means, posterior covariances, and posterior random function samples, up to a principled set of approximations, using only kernel-matrix-based linear solves. These are handled at graphics-scale with standard scalable Gaussian process methods, without involving the Poisson equation.

2. Avoid the use of mesh-based or otherwise global Poisson solves, allowing one to obtain the surface from purely-local evaluations.

3. Eliminate the coupling between the discretization structure of the enclosed bounding box and the length scale hyperparameter used in interpolation.

4. Analytically describe the Gaussian process induced by the Poisson solve, and provide user control over length scales and other hyperparameters which is retained in solution space.

## 2. Stochastic Surface Reconstruction

The goal of an *implicit surface reconstruction* algorithm is to produce a function $f : \mathbb{R}^3 \to \mathbb{R}$ representing a solid shape $\Omega \subset \mathbb{R}^3$, given data consisting of an oriented point cloud $(x_i, v_i)_{i=1}^N$, where $x_i \in \partial\Omega$ and $v_i \in \mathbb{R}^3$ are surface normals to $\partial\Omega$, both potentially noisy. To represent the surface, $f$ should take positive values inside of $\Omega$, negative values outside of it, so that the zero level set $\partial\Omega$ is the reconstructed surface. Throughout this work, we use bold italic letters such as $\boldsymbol{a}, \boldsymbol{b}$ to refer to vectors corresponding to batches of data, and bold upface letters such as $\mathbf{A}, \mathbf{B}$ to refer to corresponding matrices, with the convention that functions act on such terms component-wise.

### 2.1. Stochastic Poisson Surface Reconstruction

*Poisson surface reconstruction* is the most widely-used surface reconstruction algorithm, with numerous applications in computer graphics [28, 15, 4] and beyond [1, 12, 24]. It works as follows: first, the point cloud is interpolated onto

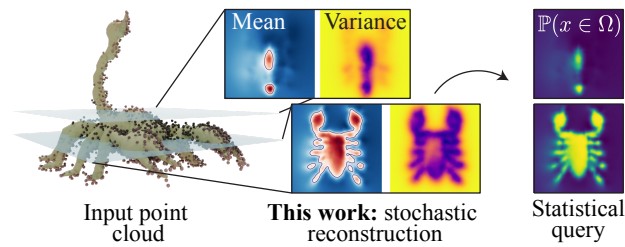

a volumetric finite element mesh to produce a vector field $v : \mathbb{R}^3 \to \mathbb{R}^3$ for which $v(x_i) \approx v_i$. Then, one reconstructs $f$, the implicit representation of $\partial\Omega$, by numerically solving the Poisson equation

$$\Delta f(x) = \nabla \cdot v(x) \tag{1}$$

over a hypercube $[0, 1]^3$, subject to Neumann boundary conditions $\nabla f \cdot n = 0$ at the hypercube's boundary, where $n$ are the normals. Given the surface is a priori unknown, the most common approach is to choose the mesh to be a regular grid, with $v$ obtained using trilinear interpolation—see Kazhdan et al. [17] for further details.

To enable the surface reconstruction algorithm to assess and propagate uncertainty for downstream use, Sellán and Jacobson [27, 26] propose *Stochastic Poisson Surface Reconstruction*: instead of a deterministic $v$, they place a Gaussian process prior $v \sim \mathrm{GP}(0, k)$ over the vector field, and calculate the respective posterior distribution in order to interpolate the point cloud data in an uncertainty-aware manner. Then, they solve a stochastic analog of (1), obtaining a random function $f \mid v$, which can be interpreted as the posterior distribution of the implicit surface given the point cloud data. This can be seen in Figure 1.

From a computational perspective, the resulting stochastic formulation is significantly more involved than ordinary Poisson reconstruction. Note that, since the Poisson equation is linear, $f \mid v$ is itself a Gaussian process—we review this and other relevant properties of Gaussian processes in the next section. Leveraging this, Sellán and Jacobson [27] propose an approach which computes the mean $\mu_{f \mid v}$ and covariance $k_{f \mid v}$ using the respective finite-element Laplacian $\mathbf{L}$ and divergence matrix $\mathbf{Z}$, as well as the kernel matrices $\mathbf{K}_{\boldsymbol{vv}}$ and $\mathbf{K}_{*\boldsymbol{v}}$ for the training and test points, respectively. To facilitate computation at point cloud sizes typical in graphics applications, Sellán and Jacobson [27] rely on the fact that $\mathbf{L}$ and $\mathbf{Z}$ are sparse, and approximate $\mathbf{K}_{\boldsymbol{vv}}^{-1} \approx \mathrm{diag}(\sigma_g \boldsymbol{\rho_v})^{-1}$, where $\sigma_g$ is a hyperparameter and $\boldsymbol{\rho}$ is a vector measuring local sampling density.

*Figure 1.* Like prior work in stochastic reconstruction [27, 26], we choose to represent the space of shapes determined by an input point cloud (left) through a volumetric Gaussian Process (middle) from which standard statistical quantities can be extracted (right).

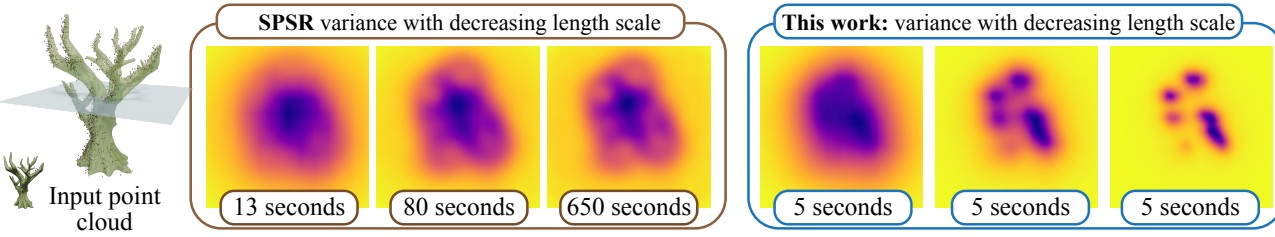

*Figure 2.* Our method removes the interpolation length scale's dependence on any grid resolution, therefore the cost to evaluate the posterior is independent of the length scale. Further, our method enables local evaluation leading to faster computations when we do not require global results. In contrast, the SPSR baseline is grid-dependent and unable to represent the correct variance at lower length scales.

The resulting method enables one to perform a number of queries that are useful in computer graphics and vision applications, like casting rays against the uncertain surface, deciding on a best next view and computing collision likelihoods. It also comes with a number of limitations. (i) One needs to solve the Poisson equation globally on the full domain, even if one is only interested in queries in a small region of space. Recent work has proposed ways to avoid this for non-stochastic Poisson surface reconstruction [8]. (ii) This solution is only obtained on a fixed grid, and one must either perform interpolation for non-grid points, or learn their values using a neural network [26], at the cost of efficiency and convergence guarantees. This artificially ties the length scale of the GP kernel to the resolution of the utilized grid—see Figure 2. (iii) The approximations used, such as for the inverse covariance, can make uncertainty behavior difficult to control. One of our goals, in this work, will be to reduce these limitations.

### 2.2. Geometric Gaussian Processes and Periodic Kernels

A *Gaussian process* $f \sim \mathrm{GP}(\mu, k)$ is a random function where, given a finite set of input points $\boldsymbol{x} = (x_1, .., x_N)$, the output $f(\boldsymbol{x}) \sim \mathrm{N}(\boldsymbol{\mu}, \mathbf{K}_{\boldsymbol{xx}})$ is multivariate normal with mean $\boldsymbol{\mu} = \mu(\boldsymbol{x})$ and covariance $\mathbf{K}_{\boldsymbol{xx}} = k(\boldsymbol{x}, \boldsymbol{x})$. A key property of Gaussian processes is that their conditional distributions $f \mid \boldsymbol{y}$, given data $f(\boldsymbol{x}) = \boldsymbol{y}$, are also Gaussian processes with explicit means and covariances. For an overview of these properties, see Rasmussen and Williams [25].

The techniques we develop will rely on ideas from *geometric Gaussian processes*—that is, Gaussian processes whose domain is not $\mathbb{R}^d$ but instead has geometric structure. Specifically, we work with Gaussian processes $f : \mathbb{T}^d \to \mathbb{R}$ over the $d$-dimensional torus $\mathbb{T}^d$. The kernels of such Gaussian processes can be viewed as periodic functions on $\mathbb{R}^d$: we thus refer to them as *periodic kernels*.

We work with the class of *periodic Matérn kernels* of Borovitskiy et al. [6], denoted $k_\nu$, $\nu \in \mathbb{R}^+$. These kernels admit analytic expressions for $d = 1$ and $\nu$ half-integer. For $\nu = 1/2$, the kernel—and a closely-related expression

$\rho_{k_\nu} : \mathbb{Z}^d \to \mathbb{R}$ called the *spectral measure* of $k$—are

$$k_{1/2}(x, x') = \cosh\left(\frac{2\pi |x - x'| - \frac{1}{2}}{\kappa}\right) \tag{2}$$

$$\rho_{k_{1/2}}(x, x') = 2\sinh\left(\frac{1}{2\kappa}\right)\left(\frac{1}{\kappa^2} + 4\pi^2 n^2\right)^{-1}. \tag{3}$$

We will make use of a number of properties of such kernels in our work. In particular, periodic Matérn kernels are *stationary*, in the sense that $k$ satisfies $k(x+c, x'+c)$ for any $x, x', c \in \mathbb{T}^d$, where addition is defined in terms of angles. Under mild regularity conditions, one can show—see for instance Borovitskiy et al. [6], Theorem 5 and Appendix B—that such $k$ admit the *Mercer's expansion*

$$k(x, x') = \sum_{n \in \mathbb{Z}^d} \rho_k(n) f_n(x) f_n(x') \tag{4}$$

where each $f_n$ is either a sine or cosine, with frequency determined by $n$. Using orthonormality of sines and cosines, can factor this expression to find an analogous representation for the Gaussian process prior $f$: this gives the *Karhunen–Loève expansion*

$$f(x) = \sum_{n \in \mathbb{Z}^d} \sqrt{\rho_k(n)} \xi_n f_n(x) \qquad \xi_n \overset{\text{iid}}{\sim} \mathrm{N}(0, 1). \tag{5}$$

For a formal treatment, see Borovitskiy et al. [6]. These expansions will play a central role in our computations.

## 3. Stochastic Poisson Surface Reconstruction

Stochastic Poisson Surface Reconstruction offers technical capabilities that go beyond the classical, deterministic Poisson reconstruction algorithm—yet, this stochasticity greatly complicates the computational pipeline needed to efficiently run it. The original work by Sellán and Jacobson [27] proposes a pipeline that mimics classical Poisson reconstruction, in the sense that once Gaussian process interpolation is performed, a traditional finite-element-based computation is used to calculate the implicit surface. While our goal will be to broadly improve the technical capabilities

of SPSR, we will start by asking: *within a Gaussian process formalism, can one calculate the implicit surface directly, without solving more than one linear system?*

### 3.1. Bypassing the Poisson Solve using Geometric Gaussian Processes

We now proceed to answer the above initial guiding question affirmatively. The mean and covariance of our Gaussian process, assuming a centered prior, are

$$\mu_{f|v}(\cdot) = \mathbf{K}_{f(\cdot)\boldsymbol{v}}(\mathbf{K}_{\boldsymbol{vv}} + \boldsymbol{\Sigma})^{-1}\boldsymbol{v} \tag{6}$$

$$k_{f|v}(\cdot,\cdot') = k(\cdot,\cdot') - \mathbf{K}_{f(\cdot)\boldsymbol{v}}(\mathbf{K}_{\boldsymbol{vv}} + \boldsymbol{\Sigma})^{-1}\mathbf{K}_{\boldsymbol{v}f(\cdot')}. \tag{7}$$

Alternatively, rather than calculate means and covariances, one can work with Monte Carlo samples from the posterior. This avoids needing to store posterior covariances, which contain $\mathcal{O}(N^2)$ entries, in memory. We can sample from the posterior using the inter-domain Gaussian process [18] form of pathwise conditioning [32, 33], which is

$$(f \mid \boldsymbol{v})(\cdot) = f(\cdot) + \mathbf{K}_{f(\cdot)\boldsymbol{v}}(\mathbf{K}_{\boldsymbol{vv}} + \boldsymbol{\Sigma})^{-1}(\boldsymbol{v} - v(\boldsymbol{x}) - \boldsymbol{\varepsilon}) \tag{8}$$

where equality holds in distribution. Note that none of the quantities arising from either the mean and covariance, or the pathwise conditioning expression, are grid-dependent. To compute the posterior, beyond efficiently solving a standard linear system of the Gaussian process type, which we will return to in the sequel, we need to compute either one or two additional kinds of expressions:

(b) The cross-covariance term $k_{f,v}(x, x')$.

(a) If computing posterior samples, joint samples of the prior vector field $v$ and its induced implicit surface $f$ defined by $\Delta f = \nabla \cdot v$.

At first, this appears challenging: both quantities we need to compute fundamentally involve the Poisson equation, which in general one must solve numerically. However, if we are able to do so, the Poisson solve is removed: it suffices to solve the random linear system once—the Poisson solve is, effectively, replaced by a grid-independent matrix multiplication.

To achieve this, the key idea will be to leverage the fact that both the Poisson equation, and appropriately-chosen Gaussian processes, are very-well-behaved in the Fourier domain [10, 6, 5, 3]. For this, we make two assumptions:

1. The prior covariance kernel $k$ is a product of stationary periodic kernels.

2. The Poisson equation defining the implicit surface has periodic boundary conditions.

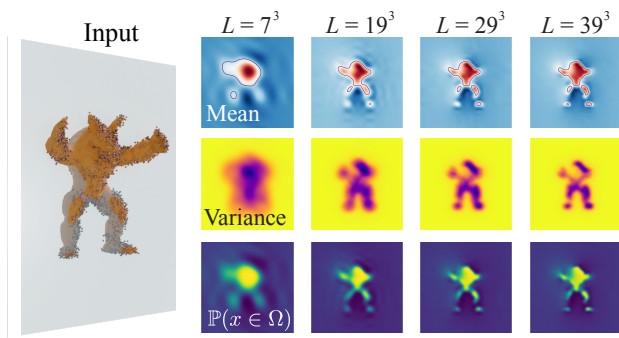

*Figure 3.* The number of Fourier basis functions used, $L$, is a key hyperparameter in our algorithm: experimentally, we find that using too few results in a loss of high frequency detail and observe diminishing returns beyond $L = 20^3$.

These assumptions enable us to apply the geometric Gaussian process machinery introduced previously: one can view both the vector field and implicit surface as functions on the torus, namely $v : \mathbb{T}^d \to \mathbb{R}^d$ and $f : \mathbb{T}^d \to \mathbb{R}$. Before proceeding, note that compared to the default setup, we have not lost anything substantive: the original Neumann boundary conditions of Kazhdan et al. [17] and Sellán and Jacobson [27] were selected primarily for convenience in the first place, and in practice one places the surface sufficiently-away from the bounding box to limit its impact.

With the Fourier domain in place, we can apply the Mercer's and Karhunen–Loève decompositions of Section 2.2 to express the cross-covariance as a Fourier series.

**Proposition 1.** *Let $v \sim \mathrm{GP}(0, k)$ where $k$ is a product of sufficiently-smooth one-dimensional stationary periodic kernels, and define $f$ by $\Delta f = \nabla \cdot v$, with periodic boundary conditions. Then*

$$k_{f,v_i}(x, x') = \sum_{\substack{n \in \mathbb{Z}^d \\ n \neq 0}} \frac{n_i \sqrt{\rho_{k_i}(n)}}{\|n\|^2} \sin(\langle n, x - x' \rangle). \tag{9}$$

All proofs are given in Appendix A. Using this expression, we can compute the cross-covariance matrix $\mathbf{K}_{f(\cdot)\boldsymbol{v}}$ approximately by truncating (9). We now show that the same techniques also allow us to efficiently jointly sample $f$ and $v$.

**Proposition 2.** *Under the same assumptions as Proposition 1, the random functions $f$ and $v$ can jointly be written*

$$v_i(\cdot) = \sum_{n \in \mathbb{Z}^d} \sqrt{\rho_{k_i}(n)} \begin{pmatrix} \xi_{i,n,1} \cos(\langle n, \cdot \rangle) \\ + \xi_{i,n,2} \sin(\langle n, \cdot \rangle) \end{pmatrix} \tag{10}$$

$$f(\cdot) = \sum_{\substack{n \in \mathbb{Z}^d \\ n \neq 0}} \sum_{i=1}^{d} \frac{n_i \sqrt{\rho_{k_i}(n)}}{\|n\|^2} \begin{pmatrix} \xi_{i,n,1} \sin(\langle n, \cdot \rangle) \\ - \xi_{i,n,2} \cos(\langle n, \cdot \rangle) \end{pmatrix} \tag{11}$$

*where $\xi_{i,n,j} \overset{\mathrm{iid}}{\sim} \mathrm{N}(0, 1)$.*

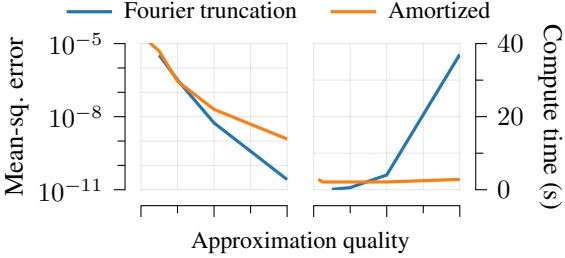

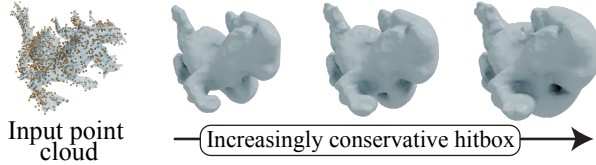

*Figure 5.* A stochastic formalism for surface reconstruction allows us to construct conservative hitboxes or cages for collision detection that account for geometric uncertainty.

*Figure 4.* We compare direct cross-covariance computation by truncating the Fourier series with its amortized analog, in terms of accuracy (kernel matrix MSE evaluated at a random batch of points) and runtime. Approximation quality is determined by number of Fourier coefficients and amortization grid size, respectively. Amortization can substantively improvement runtimes when its error level and initial training cost are acceptable.

Together, these expressions give us a way, in principle, to obtain either means and covariances, or random samples, from $f \mid v$, the latter via the pathwise conditioning expression (8)—so long as we can scalably solve the linear system, and compute all of the respective Fourier series. The computational cost of assembling the cross-covariance is $\mathcal{O}(LMN)$, and, for sampling, the cost of computing the necessary set of prior samples is $\mathcal{O}(L'NP)$, where $L$ and $L'$ are the number of Fourier basis functions used for the cross-covariance and prior, respectively, $N$ is the size of the training data, $M$ is the size of the test data, and $P$ is the number of Monte Carlo samples needed. A visualization of how the resulting approximation error affects reconstruction fidelity is given in Figure 3. Following Wilson et al. [32, 33], one can expect to need $L' \ll L$ basis functions, due to the Fourier basis' efficiency at representing random functions drawn from the prior. We will address computations costs of the cross-covariance term further in the sequel, but first proceed to handling the linear system.

### 3.2. Scalability: Training Data Size

A key challenge of working with Gaussian process methods in computer graphics applications is that they require one to solve dense linear systems, typically at cost $\mathcal{O}(N^3)$—where, in our setting, $N$ scales according to the number of points in the point cloud. While this may seem as a major challenge, observe that the (random) linear system of interest, namely

$$(\mathbf{K}_{vv} + \boldsymbol{\Sigma})^{-1}(v - v(x) - \varepsilon) \tag{12}$$

is identical in form to the linear system which appears in ordinary Gaussian process regression—and, indeed, in kernel ridge regression. The situation for computing posterior covariances is similar. This means that, even though we are in an interdomain setting which involves the Poisson equation and other machinery, we can apply standard large-scale Gaussian process techniques in an *unmodified* manner—all

of our additional machinery kicks into play only once the solve is done.

With this in mind, the question becomes how to select an appropriate linear-time Gaussian process approximation. Since correctly capturing the fine details of the object is a key goal of surface reconstruction, we restrict ourselves to approximations which are designed to perform well in *large-domain* regimes [19, 30] where the kernel's length scale is small relative to the domain covered by the data. In such regimes, inducing point methods [31, 14]—which are otherwise popular and efficient—are known to perform poorly [19, 30]. Where Cholesky factorization is not viable, we therefore apply the sampling-based *stochastic gradient descent* approach of Lin et al. [19, 20], which was shown in those works to perform well at data sizes up to $N \approx 20\text{m}$ in these regimes.

### 3.3. Amortized Cross-Covariance Computation

We now re-examine the cross-covariance computation, which costs $\mathcal{O}(LMN)$ to assemble, if computed via truncation. In contrast with the prior samples for $f$ and $v$—where, by $L^2$-optimality of the Fourier basis for representing stationary Gaussian processes, a relatively-small number of Fourier basis functions suffice [32, 33]—for the cross-covariance term, the number of Fourier basis functions $L$ needed to ensure accurate computation can become large, even in dimension three. To avoid this computation becoming the most-expensive part of the overall pipeline, we propose an amortization scheme built as follows.

Observe that, owing to stationarity, the cross-covariance $k_{f,v}(x, x')$ depends only on the distance $x - x' \in \mathbb{R}^3$. Moreover, this term does not depend on any hyperparameters except the model's length scale—and, thus, is problem-independent. Furthermore, it is smooth, since lower-frequency Fourier terms in the Mercer's expansion have larger coefficients.

Taking advantage of these observations, we propose to precompute $k_{f,v}$ on a grid in $[0, 2\pi]^d$ and apply simple linear interpolation to evaluate $k_{f,v}$ at other input locations. While this ostensibly makes our approach grid-dependent—part of what we wanted to avoid in the first place—note again

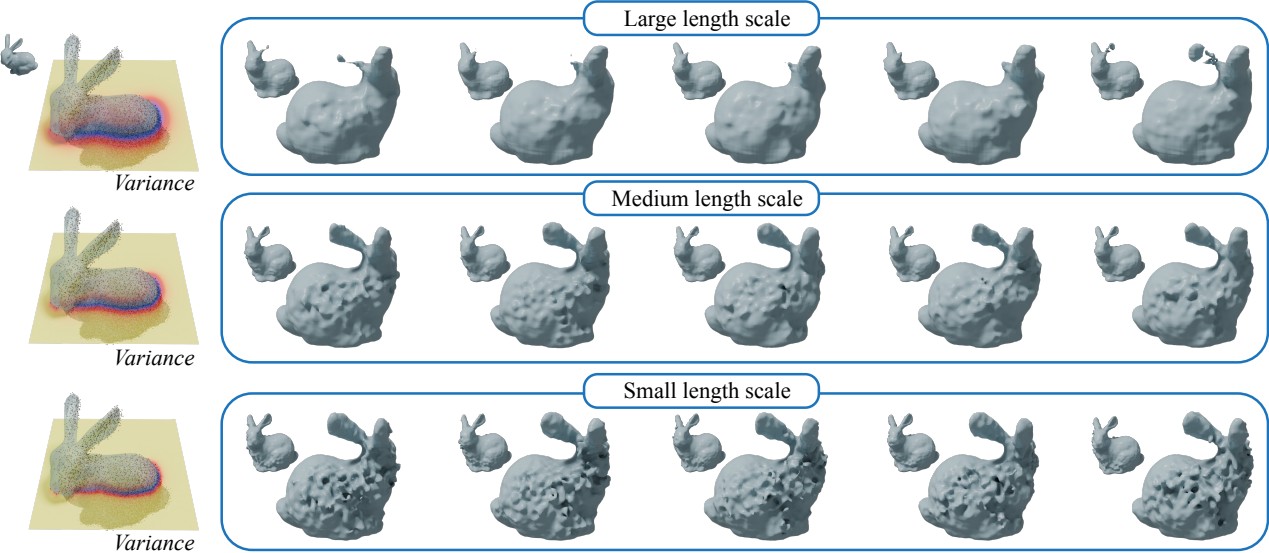

*Figure 6.* Through a pathwise-conditioning-based approach, our algorithm allows us to sample different possible surfaces from the Gaussian process determined by the input point cloud. This is to be contrasted with SPSR [27], which produces irregular samples due to various (including potentially diagonal) approximations for the covariance—a set of samples is shown in Appendix C.

that $k_{f,v}$ is smooth and problem-independent. This should be contrasted with the Poisson equation's solution, which is by definition problem-dependent and whose smoothness is completely dependent on properties of the surface being reconstructed. As consequence, one can expect grid-based amortization of $k_{f,v}$ to be substantially-less-difficult than the original, grid-based Poisson solve we sought to avoid. We examine its efficiency quantitatively in Figure 4.

### 3.4. Supported Statistical Queries

A key difference in our computational pipeline, compared to ordinary Stochastic Poisson Surface Reconstruction, is that we also allow Monte Carlo sampling, rather than only supporting computations using means and covariances. We thus detail how various posterior queries can be handled.

**Collision detection.** This involves computing the probability the reconstructed surface intersects a given known object. We handle this by sampling $M$ points on the surface of the known object, denoted $x_i$, and computing $\mathbb{P}(f(x_i) \geq 0, \forall i = 1, .., M)$ from the samples.

**Ray-casting.** This works similarly to collision detection: we compute the transmittance, which amounts to the joint probability the ray has not hit the object. Letting $x_t$ be the path of the ray, this is $\mathbb{P}(f(x_t) \leq 0, \forall t = 0, .., T)$.

**Uncertainty-aware hitbox generation.** The standard approach for generating hitboxes is to uniformly scale-up the object, then simplify its geometry, One can generalize this to an uncertainty-aware approach by scaling the object more in regions with higher uncertainty, for instance by defining the hitbox according to the implicit surface given by a function of the form $g(x) = \mu_{f|\boldsymbol{v}}(x) - \eta\sqrt{k_{f|\boldsymbol{v}}(x,x)}$, defined using the posterior mean and standard deviation. Here, for different values of $\eta$, the higher the value, the more conservative the hitbox. Extracting the zero level set as a triangle mesh can then be done using standard contouring algorithms such as marching cubes [22], as shown in Figure 5.

**Next-view planning.** A key advantage of having a probabilistic formalism is that, following approaches in areas like Bayesian optimization [11] and probabilistic numerics [13], one can use the obtained uncertainty to adaptively select where to gather data next. For scanning, Sellán and Jacobson [27] propose to adaptively select camera angles by optimizing a quantity termed the *total uncertainty*, which is defined to be the expression

$$\int_B (0.5 - |\mathbb{P}(x \in \Omega) - 0.5|)\, \mathrm{d}x. \tag{13}$$

This formulation involves computing the uncertainty over the entire volume to evaluate the score of a single ray, making it potentially expensive. Note, however, that this limitation does not affect SPSR, which requires one to query the GP over the entire volume in order to perform the Poisson solve in the first place.

Since our single-solve strategy allows us to more-efficiently query the GP a smaller set of points compared to the whole

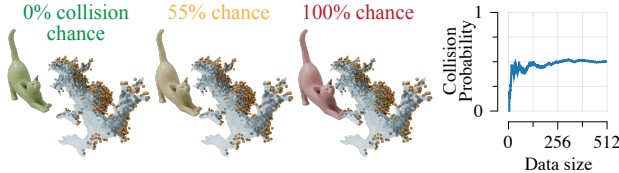

Figure 7. By querying joint probabilities from the Gaussian process produced with our algorithm, we can compute the collision probability of the three cats with the partially observed dragon.

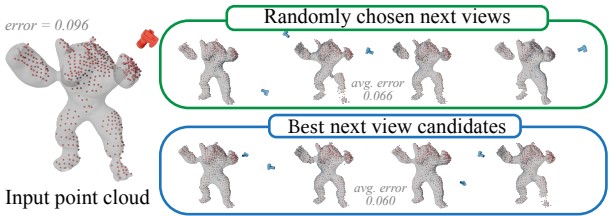

Figure 8. Our method's support for joint probability queries enable us to perform sequential scanning by adaptively selecting the next view in order to minimize total uncertainty.

volume—we show this in Figure 8, and discuss further in Section 4—we propose a modified approach which only relies on computing a function over the desired ray. For a given camera position and angle, let $x_t$ be a sequence of points along the center ray, where $t$ refers to the total travel distance. Using these points, we compute the transmittance

$$T(t) = \mathbb{P}(f(x_\tau) > 0, \forall \tau \leq t). \tag{14}$$

We then compute the total distance where the center ray is not approximately equal to zero or one, namely

$$\int_0^\infty \mathbb{1}_{\varepsilon \leq T(t) \leq 1-\varepsilon} \, \mathrm{d}t \tag{15}$$

for a given $\varepsilon > 0$. Intuitively, this measures the range of possible distances the surface can be from the ray origin.

## 4. Experiments and Applications

We now demonstrate the proposed approach empirically on a suite of example problems. Given its identical problem statement and similar theoretical backdrop, we use the original Stochastic Poisson Surface Reconstruction (SPSR) work by Sellán and Jacobson [27] as our main evaluation baseline, for which we utilize the authors' open-source implementation [29]. Implementation details and other experiment information is given in Appendix B.

### 4.1. Comparisons

To begin, we verify that our proposed algorithm output qualitatively performs at least as well as the baseline, even

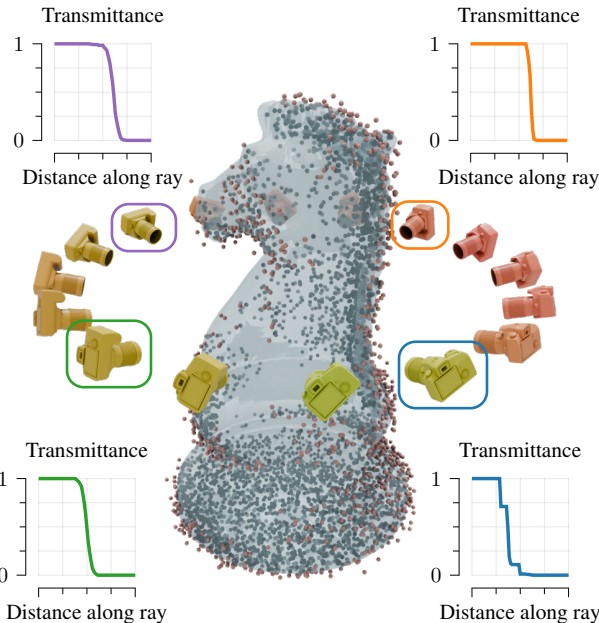

Figure 9. Our statistical formalism allows us to compute joint collision probabilities along a specific direction. In graphics, this equates to the transmittance of a cast ray, a quantity that can be used to score potential future camera positions (greener is better).

though the computational pipeline used is almost completely different. In Figure 1 and Figure 3, mirroring Figure 11 of Sellán and Jacobson [27], we see that the mean and variance produced by our approach, as well as probability queries, behave in a similar qualitative manner to those of that work.

Next, we demonstrate that our approach can provide functionality which is either not supported by SPSR, or on which SPSR scales poorly. Of the capabilities described in Section 3, we focus first on the property that it avoids the need to compute a finite element solve over a volumetric mesh or grid. A consequence of this requirement for the SPSR baseline is that the model's ability to resolve fine details is not controlled purely by the Gaussian process' length scale, and is instead also limited to the refinement level of the finite element mesh, which in turn is limited by the system's memory. Figure 2 shows that, in contrast, our model's predictions continue to become sharper as length scales decrease.

By avoiding a finite-element-based Poisson solve, our algorithm is also *output-sensitive*: Figure 11 shows that, unlike SPSR, our proposed algorithm's runtime scales with the number of queried points, and not with the size of an entire volumetric grid enclosing the point cloud. This is a critical advantage in many common high-performance computer graphics applications, like ray casting or collision detection, where the stochastic surface need only be queried at a smaller, potentially lower-dimensional number of points in

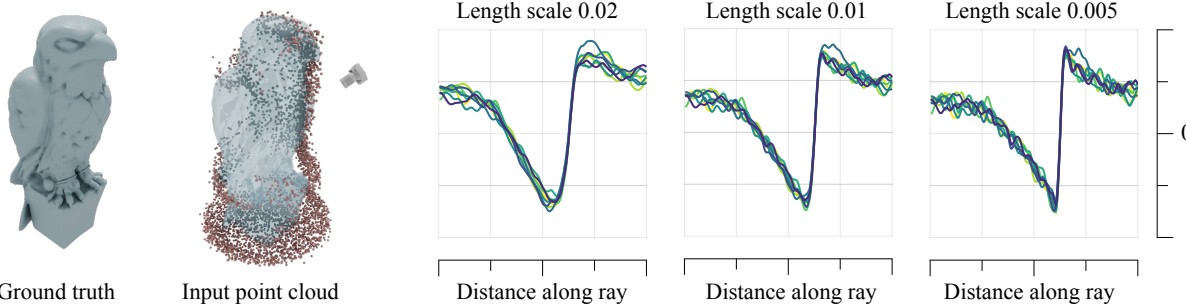

Ground truth     Input point cloud     Length scale 0.02     Length scale 0.01     Length scale 0.005

Distance along ray     Distance along ray     Distance along ray

*Figure 10.* Using our formalism, we showcase random samples drawn from $f$ at a reduced set of points—here, a ray—without the need to query the Gaussian process at a large volumetric bounding box.

space—for instance, along a line or plane, rather than over a three-dimensional volume.

Finally, both SPSR and its neural-network-based follow-up [26] discard all correlations between the input data by approximating the sample covariance with a diagonal matrix. This approximation—which is not theoretically justified beyond very specific configurations—makes SPSR unable to effectively answer statistical queries that rely on correlations. As we show in Figure 6, our approach directly generates smooth samples from the reconstructed surface, in contrast with samples arising from SPSR's diagonal approximation to the inverse kernel matrix, where smoothness comes purely from the Poisson solve performed at the end. This capability combines with output-sensitivity: from Figure 10 shows the proposed method is able to sample along a set of points given by a single ray without requiring any global Poisson-solve-like volumetric computation. While here we limit ourselves to qualitative demonstrations, to conclude, we note that handling correlations and smoothness well a critical part of principled data-acquisition algorithms such as, for instance, in Bayesian optimization: we hope the capabilities showcased potentially pave the way for similar tools in computer graphics applications.

In summary, examining the aforementioned figures, we find that our algorithm qualitatively matches the outputs produced by SPSR while alleviating a number of its limitations. We now examine how this affects downstream applications of stochastic surface reconstruction.

### 4.2. Applications

The most basic stochastic reconstruction task relevant in computer graphics, vision and robotics applications is the computation of containment queries: for example, to find if a projectile hits a target, if an autonomous car collides with an obstacle, or if a grasping pose is free of intersections. Our Gaussian process formulation allows us to answer these queries through the computation of marginal probabilities when the location is zero-dimensional—see Figure 3 and

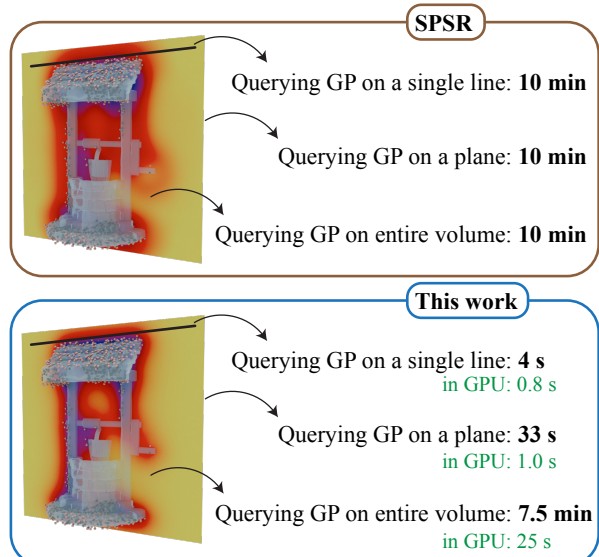

SPSR

Querying GP on a single line: **10 min**

Querying GP on a plane: **10 min**

Querying GP on entire volume: **10 min**

**This work**

Querying GP on a single line: **4 s**
in GPU: 0.8 s

Querying GP on a plane: **33 s**
in GPU: 1.0 s

Querying GP on entire volume: **7.5 min**
in GPU: 25 s

*Figure 11.* Our single-solve stochastic reconstruction algorithm is output sensitive: its complexity scales linearly with the number of test points (bottom). This is in contrast to SPSR, which requires a global volumetric FEM solve regardless of the required test set.

Figure 1—or through joint probabilities when these are one, two or three-dimensional—see Figure 7 and Figure 11. In comparison with the SPSR baseline, the output-sensitivity of our method make it ideal for containment tasks like this one, in which the captured scene—such as an entire streetscape or room—can be significantly larger than the relevant queried surface—such a car parked on the street, or a robotic gripper.

One of our method's limitations is that, even with our improved, output-sensitive performance, formally computing collision likelihoods may be beyond the computational requirements of real-time applications. Figure 5 shows that, for these cases, our method can be used to generate uncertainty-aware hitboxes for shapes in the form of triangle meshes, such that they can be efficiently tested for

intersection, using the uncertainty-aware approach outlined in Section 3.4.

In Figures 8 to 10, we show how our algorithm can be applied to the prototypical stochastic reconstruction application of ray casting and next best view planning. In particular, Figure 9, inspired by Figure 26 of Sellán and Jacobson [27], combines computing the shape's transmittance with scoring a set of potential future camera positions to select the optimal next view, outlined further in Section 3.4 for details. Unlike SPSR, using our approach to compute the score of each camera position requires querying the Gaussian process along the proposed one-dimensional camera ray, and not an entire volumetric grid surrounding the point cloud.

## 5. Conclusion

In this paper, we introduced a reformulation of stochastic Poisson surface reconstruction which requires only one linear solve, for performing Gaussian process interpolation. This is in contrast with prior work, which requires additional linear solves arising from its use of a volumetric finite element method. This was achieved by applying tools from geometric Gaussian processes in Fourier space, which effectively replaces the solve with multiplication by a matrix whose entries are given by a certain Fourier series. As a result of this formulation, the proposed method's computational costs are *output-sensitive*, in the sense that they scale with the number of points at which the posterior Gaussian process is evaluated, and not on the size of volumetric meshes or analogous quantities which depend on the space enclosing the object. The proposed method was shown to support the same set of statistical queries as prior work, and additionally provide new, random-sampling-based queries which take into account smoothness properties captured by the kernel's correlations. Our work constitutes a first step to incorporating sample-efficient data acquisition techniques from the Gaussian process and Bayesian optimization literatures into surface reconstruction and computer graphics.

## Impact Statement

This paper presents work whose goal is to apply recent advances in machine learning in order to advance the field of computer graphics. There are many potential societal consequences of our work, none which we feel must be specifically highlighted here.

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

## A. Theory

Here we prove the propositions which state the cross-covariance and prior sample expressions for the Gaussian process prior and its respective Poisson equation solution. In what follows, by *sufficiently smooth* we mean that each $k_i$ lies in the Sobolev space $H^{\frac{3}{2}+\frac{d}{2}}(\mathbb{T}^d \times \mathbb{T}^d; \mathbb{R})$: for the periodic Matérn kernels used in this work, with $\nu \geq \frac{3}{2}$, this is guaranteed by Azangulov et al. [2, 3], Theorem 20—see also De Vito et al. [9] and Borovitskiy et al. [6]. By standard stochastic PDE theory [23], this condition ensures that the Poisson equation $\Delta f = \nabla \cdot v$ admits an almost sure weak solution which is unique up to constant shifts. Taking the constant shift part to be zero, by the same theory, this solution can be viewed as a Gaussian process which is almost surely continuous. It also follows that $\rho_k$ defines a finite measure, meaning $\sum_{n \in \mathbb{Z}^d} \rho_k(n)$ is finite. We first derive the prior sampling formulas, as we will use them in the cross-covariance calculation.

**Proposition 2.** *Under the same assumptions as Proposition 1, the random functions $f$ and $v$ can jointly be written*

$$v_i(\cdot) = \sum_{n \in \mathbb{Z}^d} \sqrt{\rho_{k_i}(n)} \begin{pmatrix} \xi_{i,n,1} \cos(\langle n, \cdot \rangle) \\ + \xi_{i,n,2} \sin(\langle n, \cdot \rangle) \end{pmatrix} \tag{10}$$

$$f(\cdot) = \sum_{\substack{n \in \mathbb{Z}^d \\ n \neq 0}} \sum_{i=1}^d \frac{n_i \sqrt{\rho_{k_i}(n)}}{\|n\|^2} \begin{pmatrix} \xi_{i,n,1} \sin(\langle n, \cdot \rangle) \\ - \xi_{i,n,2} \cos(\langle n, \cdot \rangle) \end{pmatrix} \tag{11}$$

*where $\xi_{i,n,j} \overset{\text{iid}}{\sim} \mathrm{N}(0,1)$.*

*Proof.* We first describe a formal calculation which gives the desired expression, deferring a discussion on regularity to the end. Note that each $k_i$ is stationary and continuous by assumption: hence, each $v_i$ admits a Karhunen–Loève expansion

$$v_i(x) = \sum_{n \in \mathbb{Z}^d} \sqrt{\rho_{k_i}(n)}(\xi_{i,n,1} \cos(\langle n, x' \rangle) + \xi_{i,n,2} \sin(\langle n, x' \rangle)) \qquad \xi_{i,n,j} \sim \mathrm{N}(0,1) \tag{16}$$

where $\rho_{k_i}(n)$ are the respective spectral measures of the kernels $k_i$. On the other hand, note that since both the Laplacian and divergence operators are shift-equivariant, it follows that $f$ is also stationary. Since $f$ is almost surely continuous, its covariance is as well, and it admits a Karhunen–Loève expansion

$$f(x) = \sum_{n \in \mathbb{Z}^d} \sqrt{\phi(n)}(\zeta_{n,1} \cos(\langle n, x \rangle) + \zeta_{n,2} \sin(\langle n, x \rangle)) \qquad \zeta_{n,j} \sim \mathrm{N}(0,1). \tag{17}$$

By definition, $\nabla \cdot v = \Delta f$: we now formally calculate the gradient and Laplacian of the respective expansions. These are

$$(\nabla \cdot v)(x) = \sum_{n \in \mathbb{Z}^d} \sum_{i=1}^d n_i \sqrt{\rho_{k_i}(n)}(-\xi_{i,n,1} \sin(\langle n, x' \rangle) + \xi_{i,n,2} \cos(\langle n, x' \rangle)) \tag{18}$$

and

$$(\Delta f)(x) = \sum_{n \in \mathbb{Z}^d} -\|n\|^2 \sqrt{\phi(n)}(\zeta_{n,1} \cos(\langle n, x \rangle) + \zeta_{n,2} \sin(\langle n, x \rangle)) \tag{19}$$

where, since under $L^2$-regularity these expressions cannot be understood as actual random functions in the obvious sense, we defer a rigorous definition of their precise meaning to later. Continuing the calculation, since sines and cosines are linearly independent, these series must be equal term-by-term. This means

$$\sum_{i=1}^d n_i \sqrt{\rho_{k_i}(n)} \xi_{i,n,1} = \|n\|^2 \sqrt{\phi(n)} \zeta_{n,2} \qquad -\sum_{i=1}^d n_i \sqrt{\rho_{k_i}(n)} \xi_{i,n,2} = \|n\|^2 \sqrt{\phi(n)} \zeta_{n,1} \tag{20}$$

where $\phi(n)$ and $\zeta_{n,j}$ must be chosen such that the latter are IID standard normal. Choosing

$$\phi(n) = \frac{\sum_{i=1}^d n_i^2 \rho_{k_i}(n)}{\|n\|^4} \qquad \zeta_{n,1} = \frac{-\sum_{i=1}^d n_i \sqrt{\rho_{k_i}(n)} \xi_{i,n,2}}{\sqrt{\sum_{i=1}^3 n_i^2 \rho_{k_i}(n)}} \qquad \zeta_{n,2} = \frac{\sum_{i=1}^d n_i \sqrt{\rho_{k_i}(n)} \xi_{i,n,1}}{\sqrt{\sum_{i=1}^d n_i^2 \rho_{k_i}(n)}} \tag{21}$$

produces a coupling which satisfies these requirements. To complete the proof, we must give rigorous meaning to the formal computations performed above. To do so, following an approach mirroring Borovitskiy et al. [6], we reinterpret $\nabla \cdot v$ and $\Delta f$ as *generalized Gaussian fields* in the sense of Lototsky and Rozovsky [23]—that is, we interpret differentiation as acting on the respective covariance operators defined by the Mercer's expansion of both random fields. Under this interpretation, the maps $\Delta$, $\Delta^{-1}$, and $\nabla \cdot (\cdot)$ can be lifted to act on the respective covariances instead of the stochastic processes themselves: following De Vito et al. [9], when restricted to the appropriate Sobolev spaces, these maps are continuous and bijective up to appropriate constant shifts. Using this, one can calculate the generalized Gaussian field corresponding to the image of $v$ under the Poisson equation's solution map, and from it obtain a Mercer's expansion. From this, one directly obtains the respective Karhunen–Loève expansion: equating this to the expansion of $f$ above produces a calculation whose steps—up to working with modified definitions—and result match what is given above. $\square$

With these expressions derived, we are now ready to calculate the respective cross-covariance.

**Proposition 1.** *Let $v \sim \mathrm{GP}(0, k)$ where $k$ is a product of sufficiently-smooth one-dimensional stationary periodic kernels, and define $f$ by $\Delta f = \nabla \cdot v$, with periodic boundary conditions. Then*

$$k_{f, v_i}(x, x') = \sum_{\substack{n \in \mathbb{Z}^d \\ n \neq 0}} \frac{n_i \sqrt{\rho_{k_i}(n)}}{\|n\|^2} \sin(\langle n, x - x' \rangle). \tag{9}$$

*Proof.* Applying Proposition 2, this follows by direct calculation:

$$\mathrm{Cov}(f(x), v_i(x')) = \mathbb{E}(f(x) v_i(x')) \tag{22}$$

$$= \mathbb{E}\left( \sum_{n \in \mathbb{Z}^d} \sqrt{\phi(n)} (\zeta_{n,1} \cos(\langle n, x \rangle) + \zeta_{n,2} \sin(\langle n, x \rangle)) \right) \tag{23}$$

$$\times \left( \sum_{n \in \mathbb{Z}^d} \sqrt{\rho_{k_i}(n)} (\xi_{i,n,1} \cos(\langle n, x' \rangle) + \xi_{i,n,2} \sin(\langle n, x' \rangle)) \right) \tag{24}$$

$$= \mathbb{E} \sum_{n \in \mathbb{Z}^d} \sqrt{\phi(n) \rho_{k_i}(n)} (\zeta_{n,1} \cos(\langle n, x \rangle) + \zeta_{n,2} \sin(\langle n, x \rangle))(\xi_{i,n,1} \cos(\langle n, x' \rangle) + \xi_{i,n,2} \sin(\langle n, x' \rangle)) \tag{25}$$

$$= \mathbb{E} \sum_{n \in \mathbb{Z}^d} \frac{-n_i \sqrt{\rho_{k_i}(n)} \xi_{i,n,2}^2}{\|n\|^2} \cos(\langle n, x \rangle) \sin(\langle n, x' \rangle) + \frac{n_i \sqrt{\rho_{k_i}(n)} \xi_{i,n,1}^2}{\|n\|^2} \sin(\langle n, x \rangle) \cos(\langle n, x' \rangle) \tag{26}$$

$$= \sum_{n \in \mathbb{Z}^d} \frac{-n_i \sqrt{\rho_{k_i}(n)}}{\|n\|^2} \cos(\langle n, x \rangle) \sin(\langle n, x' \rangle) + \frac{n_i \sqrt{\rho_{k_i}(n)}}{\|n\|^2} \sin(\langle n, x \rangle) \cos(\langle n, x' \rangle) \tag{27}$$

$$= \sum_{n \in \mathbb{Z}^d} \frac{n_i \sqrt{\rho_{k_i}(n)}}{\|n\|^2} \sin(\langle n, x - x' \rangle) \tag{28}$$

where we pass the expectation inside the infinite series using Fubini's Theorem, for which we now verify that absolute integrability holds. To see this, first note that $\frac{n_i}{\|n\|^2} \leq 1$ and $|\sin(\cdot) \cos(\cdot')| \leq 1$, then write $\sum_{n \in \mathbb{Z}^d} \mathbb{E} \sqrt{\rho_{k_i}(n)} \xi_{i,n,j}^2 = \sum_{n \in \mathbb{Z}^d} \sqrt{\rho_{k_i}(n)} \leq \sqrt{\sum_{n \in \mathbb{Z}^d} \rho_{k_i}(n)} < \infty$, where the second-to-final step is by Jensen's inequality, and finiteness of the sum follows from the assumed smoothness of $k$. We conclude that all necessary sums and expectations are finite and do not depend on the order of integration, from which the claim follows. $\square$

## B. Experimental Details

**Implementation.** We implement our algorithm in Python using GPYTOOLBOX [29] for common geometry processing subroutines, JAX [7] for numerical computations, and render our results in Blender using BLENDERTOOLBOX [21]. All reported timings are calculated on a machine running Ubuntu 20.04 with an Intel Xeon Silver 4316 CPU, 256GB RAM, and an Nvidia RTX A6000 GPU. All of our results use the Matérn kernel with $\nu = 3/2$ and length scales between $4 \cdot 10^{-2}$ and $1 \cdot 10^{-2}$, depending on the specific mesh. Unless specified otherwise, we use $L = 100^3$ functions for the cross-covariance

and $L' = 40^3$ functions for the prior samples. The cross-covariance is amortized using a total of $50^3$ points. To generate an amortization grid, we first evenly sample $[-1, 1]$ with 50 points, raise them to the 5th power in order to concentrate values near the origin, multiply by $\pi$ to obtain points in $[-\pi, \pi]$, then take the Cartesian product for each dimension.

**Meshes.** The Armadillo, Bunny, Falcon, Scorpion, Springer, Tree, and Well meshes are from Oded Stein's repository, at: ODEDSTEIN.COM/MESHES. The Dragon mesh is originally from the Stanford 3D Scanning Repository—we use the version from Alec Jacobson's repository, at: GITHUB.COM/ALECJACOBSON/COMMON-3D-TEST-MODELS/.

**Kernel numerical stability.** The geometric Matérn-$3/2$ kernel on $\mathbb{T}^1$, given by Borovitskiy et al. [6] can be numerically unstable, particularly for small length scales, say on the order of $10^{-2}$. This occurs due to the specific form of various hyperbolic functions in the formula. To improve stability in floating point arithmetic, we equivalently rewrite the kernel as

$$k_{3/2}(x, x') = \frac{\sigma^2}{C_{3/2}} \left( \frac{\pi^2 \kappa}{3} \left( 2\kappa + \sqrt{3}\coth\left(\frac{\sqrt{3}}{2\kappa}\right) \right) \cosh(u) - \frac{2\pi^2 \kappa^2}{3} u \sinh(u) \right) \tag{29}$$

$$= \frac{\sigma^2 \pi^2 \kappa}{3 C_{3/2}} \left( \left( 2\kappa + \frac{\sqrt{3}}{\tanh\left(\frac{\sqrt{3}}{2\kappa}\right)} \right) \cosh\left(\frac{w}{\kappa}\right) - 2w \sinh\left(\frac{w}{\kappa}\right) \right) \tag{30}$$

$$= \frac{\sigma^2 \pi^2 \kappa}{3 C_{3/2}} \left( 2\kappa + \frac{\sqrt{3}}{\tanh\left(\frac{\sqrt{3}}{2\kappa}\right)} - 2w \tanh\left(\frac{w}{\kappa}\right) \right) \cosh\left(\frac{w}{\kappa}\right) \tag{31}$$

$$= \frac{\sigma^2 \pi^2 \kappa}{6 C_{3/2}} \left( 2\kappa + \frac{\sqrt{3}}{\tanh\left(\frac{\sqrt{3}}{2\kappa}\right)} - 2w \tanh\left(\frac{w}{\kappa}\right) \right) \frac{\exp\left(\frac{w - \frac{\sqrt{3}}{2}}{\kappa}\right) + \exp\left(\frac{-w - \frac{\sqrt{3}}{2}}{\kappa}\right)}{\exp\left(-\frac{\sqrt{3}}{2}\right)} \tag{32}$$

where distances run from 0 to 1, $u = \sqrt{3}\frac{|x - x'| - \frac{1}{2}}{\kappa}$, $w = \sqrt{3}|x - x'| - \frac{\sqrt{3}}{2}$, $C_{3/2}$ is a constant which ensures $k(x, x) = \sigma^2$, and $\kappa$ is the length scale parameter. Our implementation uses the unnormalized version of this expression.

## C. Additional Results

**Visualization of samples from SPSR baseline.** Here we provide a visual illustration of samples generated by the SPSR baseline, using its provided mean and covariance. We use a total of $25^3$ points, then compute the mesh via marching cubes. Given the interplay between the grid spacing and kernel length scale imposed by SPSR, this coarse grid equates to a large length scale which produces over-smoothing. In turn, unfortunately, using a finer grid—specifically, the $100^3$ grid from Figure 6—is not feasible given the memory available on our system, due to the need to form a large covariance matrix. The obtained samples are shown in Figure 12, and are seen to fail to capture a substantial portion of the reconstructed surface's fidelity due to over-smoothing. This is the case even in well-sampled regions such as the front of the surface, due to the aforementioned interaction between length scale and grid spacing. In comparison, samples obtained from the correct posterior, when computed using the techniques of our work shown in Figure 6, are qualitatively much sharper. This is in part because our approach allows for a much-larger set of $100^3$ points, which is feasible because it provides samples directly, without requiring the formation of a large covariance matrix.

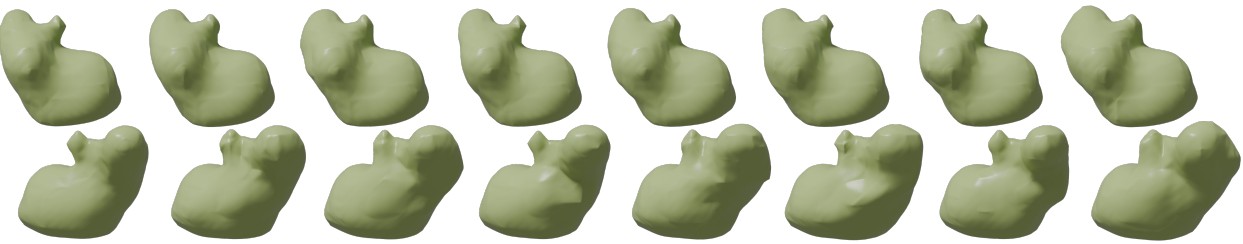

*Figure 12.* Random samples drawn using the SPSR baseline.

**Comparison of SGD vs. Cholesky.** Here we examine the behavior of using SGD for the kernel matrix solve and compare it to the standard method of using Cholesky decomposition. Throughout this work, we use the dual SGD variant proposed by Lin et al. [20]. We observe in Figure 13 that SGD is able to provide a reasonable approximation after only 10 iterations and converges to a solution not visually distinguishable from that found by Cholesky factorization within 1000 iterations. Figure 16 of the sequel additionally shows that the amount of iterations is similar across length scales—in contrast to Cholesky factorization, which will fail under sufficiently-large length scales due to ill-conditioning.

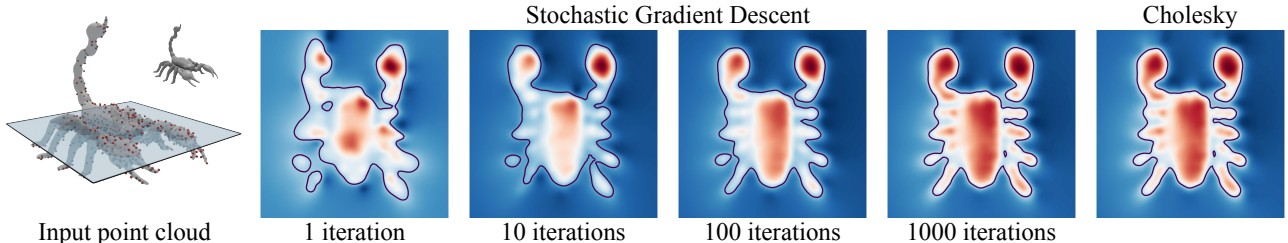

*Figure 13.* SGD converges to a qualitatively comparable solution to that found by Cholesky factorization in a few thousand iterations.

**Effect of amortization grid density.** We investigate how the grid density used for amortizing the cross-covariance function $k_{f,v}(x, x')$, using the procedure described previously in Appendix B, affects the posterior and reconstruction quality. Figure 14 shows that using an $N$ that is too small, for instance $N = 5$, results in poor performance with contain strong artifacting. On the other hand, increasing $N$ leads to higher-fidelity reconstructions with more high-frequency details.

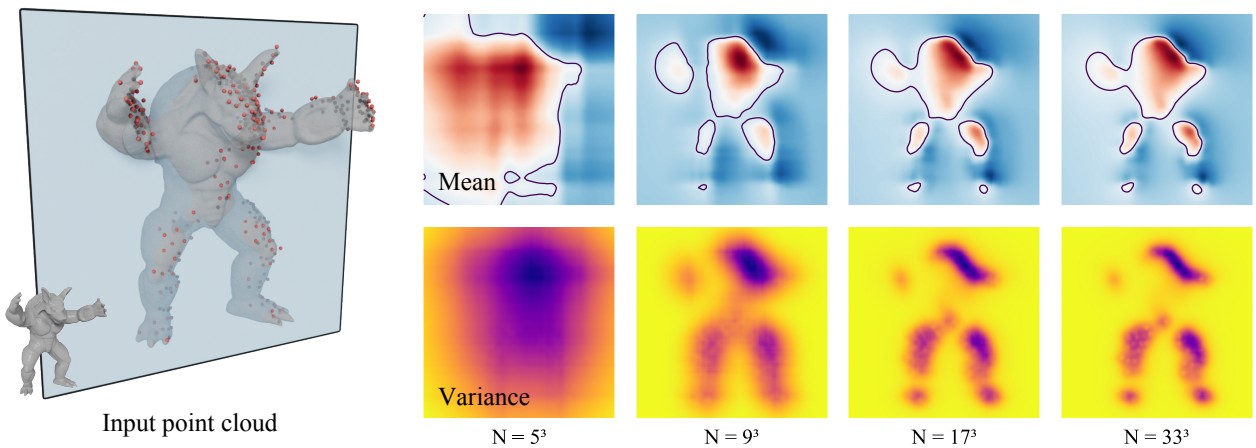

*Figure 14.* The size of the amortization grid determines the reconstruction's high-frequency level of detail.

**Runtime sensitivity to input and output size.** Figure 15 shows how our algorithm's wall-clock runtime scales with input size—that is, with the number of observed data points.

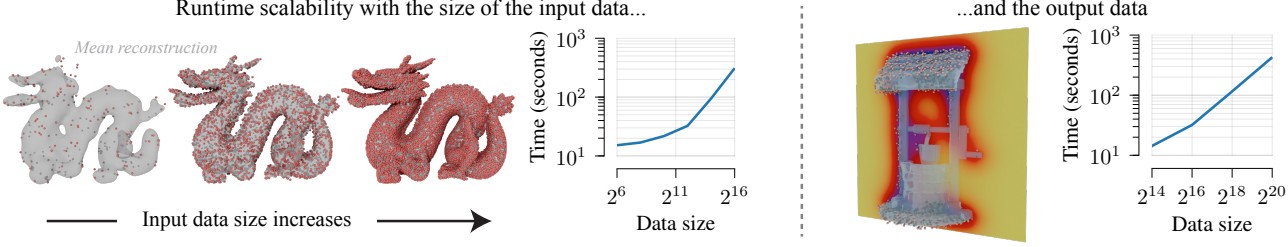

*Figure 15.* An illustration of our algorithm's input and output sensitivity.

**Runtime independence of length scale.**    SPSR, due to only being able to evaluate the posterior on a finite element mesh, couples the kernel length scale with the output density, and thus runtime as well. In this example, we attempt to match the interpolation length scale between SPSR and our method as much as possible, given their different kernels. Figure 16 gives a comparison, where each row has half the length scale compared to the row above. We observe in the first row that using large length scales with SPSR leads to too much interpolation post-solve which produces blurry results. Our method instead does not require any interpolation post-solve since by construction we can evaluate the posterior anywhere, leading to a sharper reconstruction with larger length scales. Furthermore, in practice, SPSR's runtime scales exponentially with the (inverse) length scale, in the sense that dividing the length scale by 2 increases the runtime by roughly $2^d$. On the other hand, our method's runtime is empirically constant with respect to the length scale. For comparison, in this example the amount of time to compute the mean—which is simply (a variant of) ordinary non-stochastic PSR—is $0.053$, $0.54$, and $9.1$ seconds, respectively, for the three length scales considered, in increasing order.

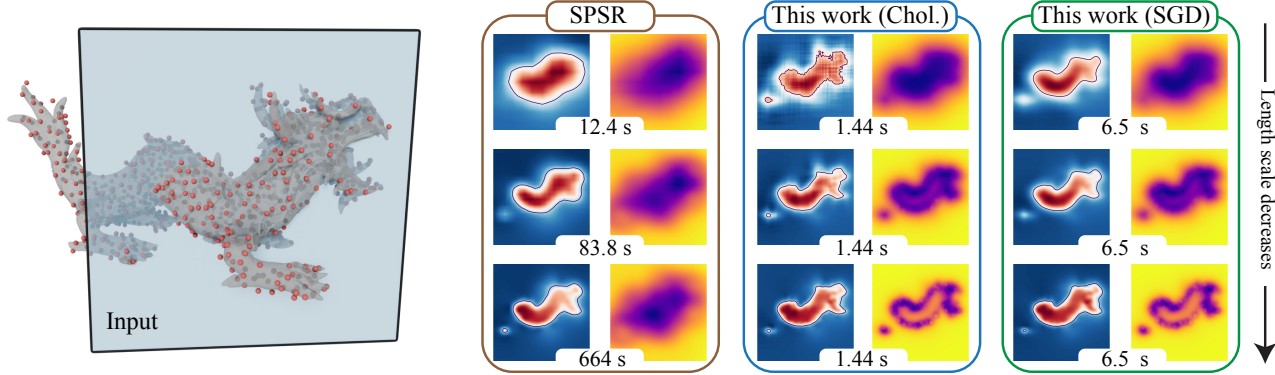

*Figure 16.* Our runtime is not sensitive to the kernel's length scale, both with Cholesky and SGD.

