# OpenReview forum: "Stochastic Poisson Surface Reconstruction with One Solve using Geometric Gaussian Processes"
_ICML.cc/2025/Conference — ICML 2025 poster_

### Official Review · Reviewer_WF3M · 2025-03-01

**Overall Recommendation:** 3

**Summary:**

The paper improves the stochastic Poisson surface reconstruction [25], which combines the interpolation and surface reconstruction into a single stage. The method avoids the complicated finite element method and makes use of Fourier transformation. It also proposes to use Monte Carlo samples from the posterior to reduce memory cost. The paper also presents several applications, e.g., collision detection and ray-casting.

## update after rebuttal
I appreciate that the authors' rebuttal addressed some of my concerns. I maintain my score, which is positive for the paper.

**Claims And Evidence:**

Fourier domain analysis relies on periodic kernel functions and boundary conditions, but actual point cloud data are often non-periodic. Does this assumption limit the practical applicability of the methods?

"one can view both the vector field and implicit surface as functions on the torus". Why torus? Is there any topology constraint? Can the method deal with high genus models?

**Essential References Not Discussed:**

N.A

**Experimental Designs Or Analyses:**

The paper improves the time and space efficiencies of the original SPSR [25]. The experiments should valid the time and space on large point cloud comprehensively. However, only a few examples are demonstrated. The numbers of points are not presented. Since approximations are used during Gaussian process, the quantitative accuracies are expected in the experiments. Currently, the evaluations cannot support the claims very well.

**Methods And Evaluation Criteria:**

The evaluation criteria in the current work are rather simplistic, and as a result, they do not provide sufficient support for effectively assessing the methods in question. I suggest to design more evaluations and compare with [25] comprehensively, including, accuracy, scalable etc.

**Other Comments Or Suggestions:**

N.A.

**Other Strengths And Weaknesses:**

N.A.

**Questions For Authors:**

When Monte Carlo sampling is used to reduce memory consumption, how is the representativeness of the sampling results to the posterior distribution ensured? Is there statistical bias due to insufficient sampling times?

It is possible to generalize the the method to screened Poisson surface reconstruction?

**Relation To Broader Scientific Literature:**

This is a very theoretical paper that uses the Gaussian process for Poisson surface reconstruction. I am not sure about the practical applicability.

**Theoretical Claims:**

The paper proposed complex theories for stochastic Poisson reconstruction. During Amortized cross-covariance, $f_{k,v}$ is pre-computed on a grid and simple linear interpolation is applied to evaluate $f_{k,v}$. Does the grid occupy too much space? What is the resolution?

---

> ### Author Rebuttal · Authors · 2025-04-01
>
> Thank you very much for your review! Let us address your points below:
>
> > "Fourier domain analysis … limit the practical applicability of the methods?”
> > “Why torus? Is there any topology constraint?”
>
> In short: **no, it does not limit applicability**. This is because the periodic boundary conditions needed to leverage our Fourier methods apply to the **bounding box inside of which the point cloud sits, not to the point cloud or reconstructed surface** - the point cloud and surface do not need to be periodic in any way.
>
> More specifically, to avoid periodic boundary conditions affecting reconstruction, we situate the point cloud sufficiently-far from the boundary to limit their effect: **this mirrors what is done in ordinary PSR**, which imposes other types of boundary conditions (say, Dirichlet or Neumann), and moves the point cloud far-enough away from the boundary so that it is largely unaffected by them.
>
> From a mathematical standpoint, the main reason for using periodic boundary conditions - and, therefore, the torus - is because (a) its Fourier analysis is much simpler compared to alternatives and (b) it works. One could instead try to develop a method using, say, properties of the sphere, but this would necessitate computations in terms of spherical harmonics, which are more complicated than the sines and cosines we use. We therefore think extensions like this are better left to future work.
>
>
> > “The evaluation criteria in the current work are rather simplistic … I suggest to design more evaluations and compare with [25] comprehensively, including, accuracy, scalable etc”
> > “Since approximations are used during Gaussian process, the quantitative accuracies are expected in the experiments.”
>
> We agree that improved evaluations would strengthen the paper (though with the caveat that we have followed the SPSR baseline paper’s lead and focused on qualitative evaluations because they are important in graphics applications), and to this end have **completed a number of additional evaluations looking at things like the effect of hyperparameters** on results, and timing experiments. We are additionally **aiming to add a quantitative next-view planning comparison** to the final manuscript draft. We describe these additional experiments to be added to the appendix - both the ones that are complete with results, and those still in progress - in detail in our response to Reviewer vrrd.
>
>
> > “Does the grid occupy too much space? What is the resolution?”
>
> We use a grid of size $50^d$. As all of the experiments are done for $d = 3$, the grid size is ~500 KB of memory. More broadly, we agree with the importance of comprehensively evaluating how grid size affects performance, and have performed **additional experiments showing a too-small amortization grid results in over-smoothing**, which we will add to next manuscript version’s appendix.
>
>
> > “The experiments should valid the time and space on large point cloud comprehensively”
>
> This is a good idea: **we have added an experiment, which uses points sampled from the Stanford Dragon mesh, and examines runtime**: we find that using 64 points takes a 15 seconds (much of it we suspect due to Python overhead), 4096 points takes about 30 seconds, and using 65536 points takes about 5 minutes. We will add a plot showing this to the next manuscript version.
>
>
> > “When Monte Carlo sampling is used to reduce memory consumption, how is the representativeness of the sampling results to the posterior distribution ensured? Is there statistical bias due to insufficient sampling times?”
>
> Since our posterior is a Gaussian process, we can use direct Monte Carlo sampling, which is by definition unbiased (as opposed to, say, other settings which require Markov chain Monte Carlo methods or similar). However, one might be concerned about variance of posterior functionals: here, it is a good question to ask how many samples are enough in practice, as is requested by Reviewer vrrd.
> To address this, we will add **additional comparisons which show how transmittance calculations and collision detection performance varies with the number of Monte Carlo samples** to the next manuscript draft.
>
>
> > “It is possible to generalize the the method to screened Poisson surface reconstruction”
>
> This is an interesting question! It seems highly plausible that an extension of our approach would work, but it would require analytically re-computing the relationship between the Karhunen-Loeve decompositions, which would be sufficiently-involved mathematically that we believe it is best to defer it to future work.
>
> ---
>
> # Summary
>
> Overall, your suggestions have led us to **improve the writing** - in particular, to emphasize that our Fourier formulation does not restrict the kinds of surfaces we can reconstruct - as well as the **strength of evaluations** used for this work. On behalf of these additions, we would gently like to ask whether you would consider increasing your score.

---

> > ### Comment · Reviewer_WF3M · 2025-04-09
> >
> > I appreciate that the rebuttal makes the work clearer. I choose to keep my positive evaluation.

---

### Official Review · Reviewer_vrrd · 2025-03-12

**Overall Recommendation:** 3

**Summary:**

The paper uses techniques from geometry Gaussian process to speed up the stochastic Poisson surface reconstruction method.

## Update after rebuttal

I appreciate the authors' efforts in providing a more nuanced discussion and additional comprehensive results. Given this, I keep my score which is already positive.

**Claims And Evidence:**

I'm not convinced by the claim that the proposed method qualitatively matches the outputs of SPSR in Section 4.1. In the first paragraph, the authors support this claim by comparing Figures 1 and 3 in this paper with Figure 11 in the SPSR paper. However, these comparisons are performed on different objects which cannot be directly compared. Since it is straightforward to generate results on the same object using identical slices, I expect a side-by-side comparison with SPSR on the same objects (including mean, variance, and probability).

**Essential References Not Discussed:**

To the best of my knowledge the essential related works are cited.

**Experimental Designs Or Analyses:**

I'm convinced by the theoretical analysis and results demonstrating the improvement in terms of the speed over SPSR, despite that, there are some limitations:

1. A key contribution of the paper is replacing the computation of means and covariances with Monte Carlo sampling from the posterior, avoiding the need to store posterior covariances. However, the paper lacks discussion on how the number of Monte Carlo samples impacts both reconstruction speed and quality.

2. A more fair qualitative comparison with SPSR is necessary to justify the quality claims, as mentioned in my comments under Claims and Evidence.

**Methods And Evaluation Criteria:**

Despite the paper’s contribution to accelerating SPSR, supported by both theoretical analysis and quantitative results, it lacks qualitative comparisons with SPSR. In contrast, SPSR provides extensive qualitative comparisons with PSR, as seen in Figures 1, 3, 7, 8, 18, and 20. These comparisons are crucial, as the contribution of speeding up SPSR is diminished if the method does not maintain the original reconstruction quality. I elaborate more on this in other sections.

**Other Comments Or Suggestions:**

-

**Other Strengths And Weaknesses:**

Overall, I acknowledge the paper's contribution to the community as a sped-up version of SPSR, with the local querying capabilities and mathematically principled approach.

**Questions For Authors:**

I would appreciate it if the authors could provide a more in-depth discussion on Monte Carlo sampling, particularly its impact on reconstruction speed and quality. Additionally, a more comprehensive qualitative comparison with SPSR would help better support the claims made in the paper.

**Relation To Broader Scientific Literature:**

The proposed method has the potential for broad applicability in accelerating various tasks, including surface reconstruction and ray casting in computer graphics, as well as collision detection in autonomous driving and human-robot interaction.

**Theoretical Claims:**

The theoretical claims look good to me.

---

> ### Author Rebuttal · Authors · 2025-04-01
>
> Thank you for your review! We appreciate that you mentioned our method **“has the potential for broad applicability”** and that we have a **“mathematically principled approach”** which was indeed part of our motivation for this work.
>
> > “I'm not convinced by the claim that the proposed method qualitatively matches the outputs of SPSR in Section 4.1”
> > “Despite the paper’s contribution to accelerating SPSR… diminished if the method does not maintain the original reconstruction quality”
>
> These are very good points: but addressing them involves some nuance. First, let us **draw attention to the following distinction between tasks** (also mentioned in our response to Reviewer otmx):
>
> 1. **“Surface reconstruction”**: given a point cloud, produce a reconstructed surface
> 2. **“Uncertainty quantification for surface reconstruction”**: given an underlying surface reconstruction algorithm, and noisy or otherwise imperfect inputs, produce an estimate of uncertainty for the reconstructed surface
>
> From this viewpoint, **our proposed algorithm’s purpose is 2 - that is, uncertainty quantification for surface reconstruction**, where the “underlying surface reconstruction algorithm” is classical PSR (or more precisely, a minor variant thereof, given our slightly-different boundary conditions).
>
> With this framing, we **agree improved evaluations would strengthen** our paper, but where the **focus should be on stochastic aspects to do with uncertainty** as opposed to the mean reconstructed surface, which is essentially the same as in classical methods. Mirroring the standard in graphics and in the SPSR paper (our baseline), our focus has been on qualitative properties, though we agree quantitative evaluations would strengthen our paper further. To this end, we have decided to add additional experiments to the next manuscript’s draft in the form of an expanded appendix:
>
> 1. This includes a **comprehensive examination of how hyperparameters affect results**: we find that (a) too-small an amortization grid density leads to over-smoothing, (b) SGD and Cholesky perform similarly (assuming the latter succeeds), (c) for sufficiently-small length scales, the SPSR baseline can result in over-smoothing, whereas our approach works well.
> 2. We have also **performed runtime comparisons**, which show our approach to be faster than SPSR as long the length scale is not too large (note that the small length scale regime is the interesting one, as this allows the algorithm to capture fine surface details).
> 3. We are additionally working on a **quantitative evaluation examining how the produced uncertainty affects next-view planning**, as a way to test how our algorithm’s numerics affect situations where one needs to use the produced uncertainty for a downstream purpose.
>
> We hope that these additions - of which the first two are complete - will help strengthen our evaluations and therefore alleviate your concerns.
>
>
> > “A key contribution of the paper is replacing the computation of means and covariances with Monte Carlo sampling from the posterior.”
> > “I would appreciate it if the authors could provide a more in-depth discussion on Monte Carlo sampling”
>
> Thank you for raising these points: due to their importance, let us respond in two parts.
>
> First, in situations where this is viable, we utilize analytic expressions derived from eqn. (6) to compute means and covariances rather than sampling: in hindsight, this point came out somewhat-hidden in our text, as we wanted to emphasize sampling as a new capability compared to baselines. We will therefore **modify the next draft to make this clearer** - thank you for drawing our attention here!
>
> Second, in situations where sampling is needed, we agree that further evaluation of the number of Monte Carlo samples needed is appropriate. Since the random functions we are sampling do come from Gaussian process posteriors, we expect the number of samples needed, in most cases, to be similar to other situations where Gaussian process sample paths are used - such questions are explored to some degree in the pathwise conditioning papers, though for different purposes. To address this, we will add **additional comparisons which show how transmittance calculations and collision detection performance varies with the number of Monte Carlo samples** to the next manuscript draft.
>
> ---
>
> # Summary
>
> Overall, your suggestions have led us to **significant improvements, both in terms clarity and especially the evaluations we present**, which we believe, based on those parts we were able to complete so far, will significantly strengthen our results. On behalf of these additions, we would gently like to ask whether you would consider increasing your score.

---

> > ### Comment · Reviewer_vrrd · 2025-04-09
> >
> > Thank you for the rebuttal. I appreciate the authors' efforts in providing a more nuanced discussion and additional comprehensive results. Given this, I keep my score which is already positive.

---

### Official Review · Reviewer_L7yZ · 2025-03-13

**Overall Recommendation:** 3

**Summary:**

Poisson surface reconstruction (Khazdan et al.) is the task of fitting a function $v(x)$ to point cloud data $(x_i, v_i)_i$ and solving $\Delta f = \nabla \cdot v$ for $f$ (subject to, e.g., Neumann boundary conditions). Then, the zero-level set of $f$ is the desired surface.
Stochastic poisson surface reconstruction (SPSR; Sellan and Jacobsen) proceeds similarly but models $v$ as a Gaussian process (GP), which makes $f \mid v$ a GP with tractable mean and covariance. But this is costly because (loosely speaking) point clouds contain many points, and because solving PDEs is expensive.
The submission proposes ways to make stochastic Poisson surface reconstruction more efficient by using geometric Gaussian processes, pathwise conditioning, and SGD-based Gaussian process training.

The paper proposes a set of refinements to make SPSR more efficient. These refinements focus on leveraging geometric Gaussian processes, pathwise conditioning, and efficient linear solvers. Specifically, the contributions include:
- Make $v$ a Gaussian process that automatically satisfies the boundary condition of the PDE. Choosing a periodic boundary implies Matern processes on the $d$-dimensional Torus do precisely that. Such processes admit known Karhunen-Loeve expansions (Borovitskiy et al.).
- Deduce a Karhunen-Loeve expansion for $f$ from $v$ and $\Delta f = \nabla \cdot v$. This is possible because operations like $\nabla$ and $\Delta$ can be computed in closed form for the Fourier-like terms in the expansions for $v$.
- Avoid working with conditional covariance matrices, and only ever sample from the posterior via pathwise conditioning (Wilson et al.), implementing joint samples from $p(f, v)$ and the cross-covariance between $f$ and $v$ via the expansions. The cross-covariance is expensive to evaluate, so an amortisation scheme is proposed. This avoids an explicit (typically, FEM-based) Poisson solve. What remains in terms of linear algebra is that pathwise conditioning requires solving a linear system involving a Gram matrix.
- To solve this linear system, the submission uses Lin et al.'s SGD-based algorithm instead of, for example, inducing points.

**Claims And Evidence:**

The submission makes the following claims (using the formulations from the paper's conclusion):
- **A single linear solve (for interpolation), as opposed to two linear solves in SPSR (one for interpolation, one for PDE solving).** This is accurate, even though the manipulation of Fourier coefficients of $f$ and $v$ could be regarded as something like a linear solve, too. (Appendix B suggests that one million summands are used for representing the cross-covariance, so evaluating the terms sounds relatively expensive, too.) But from a linear algebra perspective, there is no explicit PDE solver.
- **The computational cost of the proposed method depends on where the solver is queried, not the size of finite element meshes.** This is also accurate, even though it disregards the computational complexity of interpolation via pathwise conditioning, which solves a linear system that involves a Gram matrix with as many rows and columns as the point cloud has data points. However, this cost is the same for both SPSR and the proposal, so it's fair to ignore it.
- **The same set of statistical queries as in prior work are supported.** This is more or less accurate, because even though the queries are available, they all rely on evaluating densities, means, or standard deviations based on samples from the Gaussian, which leads to approximations. Prior work evaluates them exactly, but based on a model of reduced complexity (diagonal covariances). Qualitative results suggest that queries can be evaluated reasonably well by using samples, but there are no quantitative results (more on quantitative results in "Methods and Evaluation Criteria"). Further, I could not find information on how many samples are used for generating the queries in Figures 1 and 3, which means there are some open questions.
- **A first step in incorporating sample-efficient data acquisition schemes.** There is a mention of this point in Section 3, but no concrete suggestion and no (theoretical or empirical) evidence is provided. This seems to be more future work than a contribution (which is fair, as there is a page limit), but since both the abstract and the introduction also mention it, perhaps these claims could be weakened. For reference, the main prior work (Sellan and Jacobsen) proposes and benchmarks one such scheme (Sellan and Jacobsen, Figure 15).

In evaluating the claims, I find that the submission provides strong qualitative evidence but could benefit from additional quantitative analysis.
Overall, I think the submission's claims are supported relatively well by evidence, and even though qualitative results would make the improvements more convincing, I think this is a nice paper, and I lean towards recommending acceptance.
I discuss the lack of quantitative evidence under "Methods and Evaluation Criteria" below.

**Essential References Not Discussed:**

All essential references are discussed.

**Experimental Designs Or Analyses:**

Covered by "Methods and evaluation criteria" above.

**Methods And Evaluation Criteria:**

I appreciate the Figure's focus on readability and on visually explaining the similarities and differences between existing SPSR and the proposal. I like the numerous qualitative results. That said, the submission would be stronger with quantitative results. The absence of quantitative analysis is why I only give a borderline score.

More specifically, the experiments discuss the following scenarios:

 1. **Figures 1 and 3** show that the mean and standard deviation of the samples yield "good-looking" results. However, there is no mention of quantitative results, like calibration metrics, or at least ratios of standard deviation and mean error. It's also unclear how the reconstruction reacts to changing $L$ (or any of the other hyperparameters) beyond one sentence in the caption of Figure 3. Is there a way of taking an exactly known shape (a sphere?), generating a point cloud, and seeing how the reconstruction error is affected by parameters like $L$ or the number of data points?
 2. **Figure 2** shows how the proposed algorithm can resolve small lengthscales because it's not limited by the memory demands of a finite element mesh. It's a bit unclear what ``lengthscale'' means here. According to Section 1, it's a hyperparameter of the prior over $v$. But in Figure 2, it seems to be an (induced?) hyperparameter for $f$. I think I understand the high-level point in Figure 2, but if possible, some more precision in the terminology for "lengthscale" would be nice. And since we're talking about hyperparameters of Gaussian process models, does the proposed algorithm offer a mechanism to calibrate hyperparameters, e.g. via marginal likelihoods? And again, if there were some more quantitative error analysis, the point of SPSR not capturing these points would be more convincing than looking at a single example.
 3. **Figure 10** demonstrates that the algorithm's runtime scales with the number of query points, not the number of FEM points. This is all under the assumption that $L$ is fixed (and sufficiently large), using the amortisation from Section 3.3, and that the term $K^{-1} (\mathbf{v} - v(x))$ has been computed, correct? I am asking, because precomputing all $L$ terms and amortising the covariance feels like there is a corresponding performance gain in the FEM solver to be explored (something along the lines of precomputing the inverse of the matrix and amortising that result). I like the result in Figure 10, but perhaps there is some nuance to provide in the analysis. Or have I misunderstood something?
4. Another criticism is that the proposed algorithm contains multiple new components that are only benchmarked on surface reconstruction in combination, never on their own. Currently, it seems that the combination of approaches works well; however, for example, the combination of pathwise conditioning and SGD-based solves seems to be applicable to the baseline SPSR algorithm, too, with perhaps notable performance improvements. Same (but maybe to a lesser extent) for geometric Gaussian processes. To be fair, the paper only claims that the combination is helpful, but it would be nice to see that using either component isn't enough.




In summary, I like the demonstrations. But I think the submission would gain clarity with (some of) the following investigations:
- Investigating the role of $L$ on the reconstruction quality (on a toy example, if necessary).
- Investigating the number of samples needed for reasonable results in statistical queries. With these kinds of results, I think the submission would be strengthened
- A more quantitative version of Figure 10 (eg a "No. points queried" vs "Runtime" plot) that more clearly shows the linear complexity gain.

Ideally, there would also be independent benchmark studies for the different components. Mainly, to demonstrate whether it's the combination of geometric GPs and pathwise conditioning with SGD that leads to the good performance, or whether either of those two components suffices. That said, I understand it's a big ask, so I'm okay with this change not happening (even though I'd like to see it).

**Other Comments Or Suggestions:**

- It might help readability if the term "formal" is used more consistently. In the sentence after Equation (5), it means "rigorous". In Appendix A, it means "non-rigorous".
- It might also help readability if the term "problem-(in)dependent" is replaced by something more accurate. For example, Section 3.3 says that $k_{f, v}$ is problem-independent" which means that $k_{f, v}$ is independent of the point cloud (i.e. the surface being constructed). There are many subproblems in this algorithm (interpolation, Fourier coefficients respectively, PDE solving, amortisation, sampling, etc.), and I got confused by "problem-dependent" regularly. But this is my subjective opinion, and my recommendation doesn't depend on this change.

**Other Strengths And Weaknesses:**

See the other sections.

**Questions For Authors:**

- The proposed algorithm seems to be much faster than previous approaches to stochastic Poisson surface reconstruction. How close is it (in runtime and memory demands) to non-stochastic Poisson surface reconstruction?
- Figure 3 mentions diminishing results beyond $L=20^3$, and Appendix B mentions that the experiments use $L=100^3$. The latter is unexpected, given the former. Where does this discrepancy come from?

**Relation To Broader Scientific Literature:**

The paper extends prior work on stochastic Poisson surface reconstruction (SPSR) with a number of computational considerations. Most of these techniques are known (and cited in the paper):
- Geometric Gaussian processes
- Pathwise conditioning
- SGD-style linear-system solving

The combination of these techniques and the derivation of the Karhunen-Loeve expansion of $f$ based on that of $v$ are new, to the best of my knowledge. As such, I think the submission embeds well into related work, but also provides a series of new results.

**Theoretical Claims:**

I have checked the proofs of Propositions 1 and 2 in Appendix A. I appreciate the thorough derivation.
However, Equation (11) could benefit from more clarification.
Beyond Propositions 1 and 2, all theoretical claims are known.

---

> ### Author Rebuttal · Authors · 2025-04-01
>
> Thank you for your review! We are very happy that you recognized our approach as **“more efficient”** than prior ones and that our **“claims are supported relatively well by evidence.”** We address your questions below:
>
> > “manipulation of Fourier coefficients … something like a linear solve.”
> > “Appendix B suggests … relatively expensive”
>
> This is a fair point - but let us add a critical distinction: **our Fourier coefficients depend only on the kernel and not on the point cloud,** unlike the linear solves in SPSR. This property enables amortization, so one can precompute Fourier-coefficient manipulations once in advance, rather than on a per-point-cloud basis.
>
> > “how many samples … generating the queries in Figures 1 and 3”
>
> These are done using analytic expressions derived from eqn. (6) rather than sampling: in hindsight, this point came out somewhat-hidden in our text, as we wanted to emphasize sampling as a new capability compared to baselines. We will modify the next draft to improve this.
>
> > “A first step in incorporating … one such scheme (Sellan and Jacobsen, Figure 15).”
>
> We largely agree, with one tiny exception: **we do introduce a modified score function in Sec. 3.4**, which can be evaluated much faster than the one from the original SPSR paper, but is otherwise similar. Due to space constraints, we cut this in favor of future work: for completeness, we will **add a comparison between the two in the appendix of the next draft**, following the setup presented in Figure 15 of "Neural Stochastic Screened Poisson Reconstruction", Sellán and Jacobson, SIGGRAPH Asia 2023 (which studies the more-general screened Poisson surface reconstruction setting), and will modify our claims as you suggest.
>
> > “could benefit from additional quantitative analysis”
>
> We acknowledge that our results are less-quantitative than many ML works: this is similar to prior work such as SPSR, and reflects norms in the graphics community. We nonetheless agree quantitative evals would strengthen the work, and will add a number of such comparisons to the appendix, described in further detail in our response to Reviewer vrrd.
>
> > “…unclear how the reconstruction reacts to changing $L$ … Figure 3”
> > “…role of $L$ on the reconstruction quality”
>
> Thank you for this idea. We’ve done some preliminary tests, and **found that setting $L$ too small results in loss of high-frequency details**. The same holds for using too-coarse a grid for amortization. We will add this to the next draft’s appendix.
>
> > “Figure 2 shows … (induced?) hyperparameter for “f”.”
>
> In both cases, **this is a hyperparameter**, specifically the number $\kappa$ distances are scaled by before going into the kernel, $k(x,x') = k(\frac{x-x'}{\kappa})$, which determines both $v$ and in turn $f$. We will make this clearer.
>
> > “… mechanism to calibrate hyperparameters … more convincing”
>
> This is an excellent question. Since our kernel-matrix solve is identical to that of an ordinary GP, one way to do this would be to apply **standard maximum marginal likelihood techniques**. We will add discussion on this.
>
> > “Figure 10 … number of query points … Or have I misunderstood something?”
>
> You are correct that $L$ is fixed and sufficiently large (we use $L = 100^3$ as mentioned in Appendix B). Here, allow us to again emphasize that our cross-covariance is independent of the input point cloud. While one may be able to optimize FEM meshes for specific point clouds, **we can compute our cross covariance once and reuse it for any point cloud.**
>
> > “Another criticism is that the proposed… either component isn't enough” and “Independent benchmark studies”
>
> This is a good idea. We ran additional comparisons on the difference between SGD and Cholesky factorization, which **show that performance (provided Cholesky works) is comparable**, and will add them to the appendix. In the case of pathwise conditioning, we mainly view this as new functionality, since SPSR’s global nature limits the attainable resolution of samples.
>
> > “A more quantitative version… linear complexity gain”
>
> Good idea! We will add this to the next draft’s appendix.
>
> > “Equation (11) .. clarification”
>
> Here, $P(x \in \Omega) = P(f(x) \leq 0)$ as our implicit surface representation takes on negative values inside $\Omega$, zero on the boundary of $\Omega$, and positive values outside of $\Omega$. We will make this more clear in the camera ready.
>
> > “term "formal" .. consistently”
> > “term "problem-(in)dependent"
>
> Thank you for these - we will fix them.
>
> > “How close is it … to non-stochastic [PSR]?”
>
> This is a good point and might help readers get a sense of how expensive the method is  - we will add numbers to the next draft.
>
> ---
>
> # Summary
>
> Your suggestions have led to **improved clarity** (via the many points - thank you for them!), as well as **strength of evaluations** of this work. Given these additions and above clarifications, we would gently like to ask whether you would consider increasing your score.

---

> > ### Comment · Reviewer_L7yZ · 2025-04-03
> >
> > Thanks for the detailed rebuttal! I appreciate that the authors ran additional experiments. That said, I'll keep my "weak accept" assessment.
> >
> > I still find the paper interesting and the technique promising, but the lack of quantitative results still affects clarity. The rebuttal mentions further experiments in line with my suggestions but only provides rough summaries, not concrete setups or results -- just a promise to include numbers in the next draft. Including tables or figures in the rebuttal would have made it easier to assess the changes.
> >
> > Again, I thank the authors for the rebuttal. I still like the submission and will keep my (already positive) score.

---

> > > ### Author Response · Authors · 2025-04-05
> > >
> > > Thank you! Unfortunately, as we were character-limited in our response above, we had to keep the description of the additional comparisons we have performed brief. Here's a more-detailed description of the additional experiments and preliminary results:
> > >
> > > 1. **SGD vs. Cholesky** [complete]:
> > > We took a scene consisting of the Scorpion mesh, then compute the posterior using both SGD and Cholesky factorization for the GP solve. We found that SGD typically converges to the solution found by Cholesky in about 1000 iterations. We also found, using a length scale that is too large (but not so large that factorization outright fails), that Cholesky factorization can produce poor-quality solutions which do not visually resemble the mesh. On the other hand, SGD is still able to find good solutions in this regime, and is less sensitive to the precise length scale value used for larger length scales.
> > > 2. **Amortization Grid Density** [complete]:
> > > We evaluated reconstruction of the Armadillo mesh, using a titan of 5^3, 9^3, 17^3, and 33^3 points for the amortization grid density (following Sec. 3), and compute the posterior under each one. We find, visually, that using smaller grid densities leads to a loss of high-frequency details in both the reconstruction and in the variance used to represent uncertainty.
> > > 3. **Runtime Comparisons** [complete for input sensitivity]:
> > > We measure how long both our algorithm and the SPSR baseline take, on a wall-clock basis,  for performing reconstruction on the Stanford Dragon mesh as a function of the number of input points. For the former, we find that using 64 points takes 15 seconds (much of it we suspect due to Python overhead), 4096 points takes about 30 seconds, and using 65536 points takes about 5 minutes. We will add a plot showing these, and an additional comparison involving the number of output points.
> > > 4. **Next View Planning** [to be added]:
> > > We will simulate progressive scans using both the camera score introduced by the SPSR paper, and the one we introduce in Sec. 3.4. At each time, we will take the mean reconstruction, display it, and additionally provide a quantitative comparison using the Chamfer distance to the ground truth mesh.

---

### Official Review · Reviewer_otmx · 2025-03-14

**Overall Recommendation:** 3

**Summary:**

In the paper, the authors reformulated the stochastic Poisson surface reconstruction by introducing geometric Gaussian processes and periodic kernels. Their proposed method achieves similar results while addressing a number of limitations to increase computational efficiency.

**Claims And Evidence:**

The claims made in the submission should have been supported by evidence.

**Essential References Not Discussed:**

All essential references should have been discussed.

**Experimental Designs Or Analyses:**

Overall, the experimental designs and analyses are sound.

**Methods And Evaluation Criteria:**

Not sure. As a paper introducing a new Poisson surface reconstruction method, there are no quantitative evaluations and comparisons of reconstruction quality.

**Other Comments Or Suggestions:**

N/A

**Other Strengths And Weaknesses:**

I find little weakness in the paper overall. It is appreciated that the authors also demonstrate various applications of their method. However, as a method focused on reconstructing surfaces from point clouds, the paper does not directly compare the reconstruction quality and runtime with other approaches. Additionally, the paper only compares its approach with a single baseline, which appears to be somewhat limited. It fails to provide a clearer understanding of the method's efficiency and accuracy relative to existing techniques.

**Questions For Authors:**

N/A

**Relation To Broader Scientific Literature:**

The paper introduces an advanced surface reconstruction methodology.

**Theoretical Claims:**

The proof for theoretical claims should be correct.

---

> ### Author Rebuttal · Authors · 2025-04-01
>
> Thank you for your review! We appreciate your recognition that our approach **“increase[s] computational efficiency”** and that our paper has **“little weakness”** and are **delighted by these comments**! We address your key comments below:
>
> > “no quantitative evaluations and comparisons of reconstruction quality”
> > “However, as a method focused on reconstructing surfaces from point clouds, the paper does not directly compare the reconstruction quality and runtime with other approaches”
> > “Additionally, the paper only compares its approach with a single baseline, which appears to be somewhat limited”
>
> These comments, while brief, are also quite important: let us address them in some detail. First, let us **draw attention to the following subtle distinction between tasks**:
>
> 1. **“Surface reconstruction”**: given a point cloud, produce a reconstructed surface
> 2. **“Uncertainty quantification for surface reconstruction”**: given an underlying surface reconstruction algorithm, and noisy or otherwise imperfect inputs, produce an estimate of uncertainty for the reconstructed surface
>
> From this viewpoint, **our proposed algorithm’s purpose is 2 - that is, uncertainty quantification for surface reconstruction**, where the “underlying surface reconstruction algorithm” is classical PSR (or more precisely, a minor variant thereof, given our slightly-different boundary conditions).
>
> As such - and given that algorithms of this kind are rather new - **SPSR is the only appropriate baseline we are aware of**. Mirroring the standard used in that work - and more broadly in graphics - our results focus on qualitative behavior, though we agree more quantitative comparisons would make the paper stronger.
>
> To achieve this, in addition to the quantitative results that we currently have - such as runtime performance benchmarks - we have **performed additional evaluations such as hyperparameter comparisons** (as requested by other reviewers and described in our responses there). In addition, inspired by the need to quantitatively evaluate uncertainty in a manner fitting the above framing, we will also **add a quantitative benchmark of our next-view planning heuristic** (compared to SPSR baseline), following the setup in Figure 15 of "Neural Stochastic Screened Poisson Reconstruction", Sellán and Jacobson, SIGGRAPH Asia 2023 (which studies the more-general screened Poisson surface reconstruction setting). We anticipate to complete this by the next manuscript draft.
>
> ---
>
> # Summary
>
> Overall, your suggestions have led us to **significant improvements, especially in clarifying appropriate positioning** for this paper, but also in the right way to handle quantitative comparisons. On behalf of these additions - including extra experimental results described in our responses to other referees, which were inspired in part by points that became apparent to us through your review - we would gently like to ask whether you would consider increasing your score.

---

### Decision · Program_Chairs · 2025-05-01

**Decision:**

Accept (poster)

**Comment:**

The paper introduces a more efficient variant of Poisson Surface Reconstruction. It was well-received by the reviewers, converging to unanimous 4xWA. The evaluation was already positive pre-rebuttal, and mentioned concerns were mostly regarding clarity in writing, which have been addressed by the authors in the rebuttal, according to the reviewers.
I congratulate the authors to the great work and follow with an accept recommendation accordingly.